# Pulsed-electromagnetic-field induced osteoblast differentiation requires activation of genes downstream of adenosine receptors A2A and A3

**Niladri S. Kar**[1], **Daniel Ferguson**[1¤], **Nianli Zhang**[2], **Erik I. Waldorff**[2], **James T. Ryaby**[2], **Joseph A. DiDonato**[1]*

**1** Department of Cardiovascular & Metabolic Sciences, Lerner Research Institute, Cleveland Clinic, Cleveland, OH, United States of America, **2** Orthofix, Inc., Lewisville, TX, United States of America

¤ Current address: School of Medicine, Washington University at St. Louis, St. Louis, MO, United States of America

* DIDONAJ@ccf.org

**Data Availability Statement:** All relevant data are within the manuscript and its Supporting Information files.

## Abstract

Pulsed-electromagnetic-field (PEMF) treatment was found to enhance cellular differentiation of the mouse preosteoblast, MC3T3-E1, to a more osteoblastic phenotype. Differentiation genes such as *Alp*, *BSPI*, *cFos*, *Ibsp*, *Osteocalcin*, *Pthr1* and *Runx2* showed increased expression in response to PEMF stimulation. Detailed molecular mechanisms linking PEMF to the activation of these genes are limited. Two adenosine receptors known to be modulated in response to PEMF, *Adora2A* and *Adora3*, were functionally impaired by CRISPR-Cas9-mediated gene disruption, and the consequences of which were studied in the context of PEMF-mediated osteoblastic differentiation. Disruption of *Adora2A* resulted in a delay of *Alp* mRNA expression, but not alkaline phosphatase protein expression, which was similar to that found in wild type cells. However, *Adora3* disruption resulted in significantly reduced responses at both the alkaline phosphatase mRNA and protein levels throughout the PEMF stimulation period. Defects observed in response to PEMF were mirrored using a chemically defined growth and differentiation-inducing media (DM). Moreover, in cells with *Adora2A* disruption, gene expression profiles showed a blunted response in *cFos* and *Pthr1* to PEMF treatment; whereas cells with *Adora3* disruption had mostly blunted responses in *AlpI*, *BSPI*, *Ibsp*, *Osteocalcin* and *Sp7* gene activation. To demonstrate specificity for *Adora3* function, the *Adora3* open reading frame was inserted into the *ROSA26* locus in *Adora3* disrupted cells culminating in rescued PEMF responsiveness and thereby eliminating the possibility of off-target effects. These results lead us to propose that there are complementary and parallel positive roles for adenosine receptor A$_{2A}$ and A$_3$ in PEMF-mediated osteoblast differentiation.

**Funding:** JAD was funded by a sponsored research grant from Orthofix (https://www.orthofix.com/). NSK also received salary support from this grant. Orthofix also provided support for this study in the form of salaries for NZ, EIW, and JTR. The specific roles of these authors are articulated in the 'author contributions' section. No additional external funding was received for this study.

**Competing interests:** The authors have read the journal's policy and have the following competing interests: NZ is an employee of and own stock in Orthofix Medical Inc. and EIW and JTR are consultants and own stock in Orthofix Medical Inc. (https://www.orthofix.com/). There are no patents, products in development associated with this research to declare. The pulsed electromagnetic signal (PhysioStim®, Orthofix Medical Inc., Lewisville TX) used in this study is FDA-approved for the clinical treatment of long bone non-unions. This does not alter our adherence to PLOS ONE policies on sharing data and materials.

**Abbreviations:** DM, Differentiation Medium; PEMF, Pulsed-electromagnetic-field; qRT-PCR, Quantitative Reverse Transcription-Polymerase Chain Reaction.

# Introduction

Bone fracture is the most common musculoskeletal injury, and healing of such injury involves complex cellular and molecular processes [1,2]. Fracture healing is a multistep biological process, involving temporally defined stages including hematoma, inflammation, fibro-vasculature, bone formation and remodeling. Intermittent pulsed-electromagnetic-field (PEMF) stimulation has long been reported to heal bone fractures, especially delayed and non-unions [3–9]. PEMF, a noninvasive and safe procedure, has received extensive attention in clinical applications over last two decades. However, the biological mechanisms behind its success remain mostly enigmatic. Proliferation of different cell types, including neural stem cells [10], bone marrow mesenchymal stem cells [11], intervertebral disc cells [12], tendon cells [13], myoblasts [14], and finally osteoblasts [15] have been reported to be enhanced by PEMF stimulation. Reinforcement of intracellular calcium transients [15] and activation of the mammalian target of rapamycin (mTOR) signaling pathway leading to proliferation [16] of osteoblast cells have been described as possible effector pathways activated through intermittent treatment with electromagnetic pulses. PEMF has been reported to also modulate cAMP levels [17] and thus to induce an anti-inflammatory response in a variety of cell types [18–21].

Adenosine receptors (ADRs) have been shown to play a crucial role in both the anti-inflammatory response and in bone formation. ADR $A_{2A}$ has been reported to regulate anti-inflammatory effects, bone metabolism and hemostasis, increase cAMP levels through activation of adenylcyclase and Akt-mediated nuclear localization of the transcription factor β-catenin [22–25]. ADR $A_3$ has been shown to regulate inflammatory responses and to decrease adenylcyclase activity leading to lower cAMP levels [26]. However, the role of ADR $A_3$ in bone formation is largely unknown although it is expressed on osteoblasts and osteoblast precursors [27] and $A_3$ selective agonists have been shown to stimulate cell proliferation while $A_3$ inhibitors blocked this effect [28]. Interestingly, PEMF has been known to increase plasma-membrane densities of $A_{2A}$ and $A_3$ [22,29], the physiological implications of which relative to pre-osteoblast to osteoblast differentiation is unclear and is a driving force for our current study.

Though the enhanced healing of non-union bone fractures by PEMF-mediated osteoblast proliferation and differentiation has been documented for decades, leading to an FDA approval of PEMF devices [30], the cellular and molecular mechanisms leading to such biological effects have not been well-defined in terms of their role in PEMF-mediated fracture healing. Here, we propose that the Adenosine Receptors $A_{2A}$ and $A_3$ play positive roles of PEMF-mediated osteoblast differentiation and provide evidence of the operative molecular mechanisms each contributes to.

# Materials and methods

## Plasmids and stable lines

Homology Directed Repair (HDR) mediated CRISPR-Cas9 gene disruption technique (Origene Technologies, Inc.) was used to gene knock-out Adora $A_{2A}$ and $A_3$, according to the manufacturer's protocol. Briefly, in this two-plasmid system, cells were transfected with a plasmid harboring Cas9 and gRNA specific for early in the first exon of the target genes and one with a cassette expressing antibiotic marker genes (blasticidin for $A_{2A}$ and puromycin for $A_3$) flanked by homology arms for the target genes. After seven passages, cells were treated with antibiotics and individual clones were isolated, expanded and PCR screened for integration.

For inducible gene knock-in, the cumate inducible expression system was used (System Biosciences, LLC) with modification. Briefly, to make the donor plasmid, the $A_3$ open reading frame (ORF) was subcloned in the SparQ-T2A plasmid downstream of the cumate promoter,

and left and right homology arms for mouse ROSA26 locus were synthesized from GenScript and were subcloned upstream of cumate promoter and downstream of SV40 polyA sequence, respectively. pX330-Cas9-ROSA26gRNA was made by subcloning ROSA26gRNA sequence in pX330-Cas9 (Addgene #42230). This donor plasmid together with Cas9-Rosa26gRNA plasmid were co-transfected into $A_3$ disrupted cells, and stable cloned were isolated and screened, and the best expressing clone under cumate treatment was used for further experiments.

## Cell culture

Mouse preosteoblast cell MC3T3-E1 was bought from ATCC and grown in the growth medium (GM), made of the MEM-α without ascorbic acid supplemented with 10% fetal bovine serum and penicillin-streptomycin. The GM additionally supplemented with 50 μg/ml ascorbic acid and 10 mM β-glycerophosphate was used as a chemically defined differentiation medium (DM) [31].

## PEMF treatment

The PEMF signal used in this study had similar waveform characteristics to the clinically approved PhysioStim® (Orthofix Medical Inc., Lewisville TX) signal used to heal long-bone non-unions. The signal has a pulse frequency of 3.85 kHz and slew-rate of 10 T/s (30). Cells were cultured in GM, and exposed to daily 4 hours PEMF treatment for up to 16 days.

## Quantitative RT-PCR

Osteoblast differentiation was measured by activation of genes, *Alp*, *cFos*, *Ibsp (Bone Sialoprotein II, BSPII)*, *Osteocalcin (BGLP)*, *Osteopontin (Bone Sialoprotein I, BSPI, SPPI)*, *Pthr1*, *Runx2* and *Sp7*, with 4 hours daily PEMF treatment, followed by total RNA extraction using TRIzol reagent (Ambion/Life Technologies) following the manufacturer's protocol. First-stand cDNAs were synthesized from mRNAs by reverse transcription using Superscript IV (Invitrogen) as per manufacturer's protocol and expression levels were assayed by quantitative RT-PCR with Taqman Gene Expression (Applied Biosystems) primers-probes (AlpI: Mm00475834_m1; BSPI/SPP1/Osteopontin: Mm00436767_m1; Ibsp/BSPII: Mm0049255_m1; cFos: Mm00487425_m1; Osteocalcin/Bglap: Mm03413826_mH; Pthr1: Mm00441046_m1; Runx2: Mm00501584_m1; Sp7/Osterix: Mm04209856_m1) in a StepOne Plus (Applied Biosystems) real-time PCR instrument. Each data set was normalized by untreated cells grown for 8 days, and GAPDH (Mm99999915_g1) was used as housekeeping gene. Each experiment was performed in biological triplicates and qRT-PCRs were performed for each in duplicates.

## Alkaline phosphatase assays and Alizarin Red staining

The differentiation process was also monitored by induction of alkaline phosphatase enzymatic activity in cell lysates and by cell staining every 4 days up to 16 days. For alkaline phosphatase assay, cells were lysed in homogenizer (Fisherbrand Pellet Pestle Microtubes), aliquots were incubated with assay buffer containing pNPP at room temperature for 30 mins, and $OD_{405}$ was read [31,32]. For alkaline phosphatase staining, cells were fixed for 30 sec with 4% paraformaldehyde, washed and incubated with alkaline-dye mixture containing 0.01% naphthol AS-MX phosphate and 0.24 mg/ml Fast Blue RR at room temperature for 30 min, washed and photographed.

For mineralization assay, Alizarin Red staining was performed at similar time points as above [31]. Briefly, cells were fixed with 4% paraformaldehyde, washed and incubated with 2% Alizarin red for 5 minutes at room temperature, washed and photographed.

## Western Blot

To examine at the effect on protein expression, cells were treated with PEMF (4 hours daily) or DM for 12 days, whole-cell proteins were extracted using lysis buffer containing 1% NP-40 and 0.5% TX-100, and protein concentration determined with DC Protein Assay (Bio-Rad). Total cell proteins (50 μg/lane) were fractionated on denaturing, reducing 10% or 16% SDS-PAGE and transferred onto PVDF membranes. The membranes were blocked with 4% Blotting-grade Blocker (Bio-Rad) in PBS and probed for one hour at room temperature with 1:1000 dilutions (in PBS) of antibodies against the following proteins: BSPI (PA5-79423, Invitrogen), IBSP (5468S, Cell Signaling) and Osteocalcin (ab93876, Abcam) with GAPDH (MA5-15738, Invitrogen) serving as loading control. Blots were washed three times 5 minutes each with PBS containing 0.05% Tween-20 and then incubated with 1:10000 dilution of secondary antibodies (HRP-conjugated) anti-mouse (7076S, Cell Signaling) or ant-rabbit (7074S, Cell Signaling), for 1 hour at room temperature. Finally, the blots were washed 3 times by 5 minutes each, before being developed with ECL Prime Western Blotting Detection reagent (RPN2232, GE Healthcare) per manufacturer's direction.

## Images and statistical analysis

All experiments were performed at least in biological triplicate. The Student's two-tailed T-test was used to analyze experimental vs control cell types or treatment conditions, and a σ value of $p < 0.05$ denoted with asterisks to emphasis significance.

## Results

### PEMF stimulates osteoblast differentiation

We examined the effect that PEMF elicited on the expression of select genes known to be associated with the early stages of pre-osteoblast to osteoblast differentiation. Pre-osteoblast MC3T3-E1 cells were treated with PEMF with 4 hours daily cycles, or differentiation media (DM) for up to 16 days. Cell proteins were extracted and assayed for enhanced alkaline phosphate enzymatic activity normalized to cell protein, a typical hallmark of osteoblast differentiation, and significant activation was detectable due to both treatments (**Fig 1A**). Apart from *Alp*, the other osteoblast differentiation genes, viz. *BSPI*, *cFos*, *Ibsp*, *Osteocalcin*, *Pthr1*, and *Sp7*, were also found to have significantly increased expression in response to PEMF treatment as measured by qRT-PCR from total RNA normalized to glyceraldehyde dehydrogenase (GAPDH) (**Fig 1B**). *Alp*, *BSPI*, *Ibsp* and *Osteocalcin* are the top four most prominent responding genes both to PEMF and DM treatments under these experimental conditions. PEMF was also found to increase alkaline phosphate cell staining, and Alizarin Red staining for mineralization in a dose-dependent manner (**Fig 1C**). These results further confirmed that PEMF stimulates osteoblastic differentiation of MC3T3-E1 under the experimental conditions employed here.

### *Adora3* disruption decreases alkaline phosphatase activation by PEMF in osteoblast differentiation

To investigate the role of Adenosine receptors $A_{2A}$ and $A_3$, we began by disrupting the expression of these two genes individually or in combination in MC3T3-E1 cells employing a homology directed recombination (HDR)-based CRISPR-Cas9 methodology strategy (see **Fig 2**) and then exposing the wild-type (WT), single disrupted and doubly disrupted cellular clones to PEMF (4 hr/day) for the indicated time periods. The PCR screening results confirmed that disrupted clones were homologously disrupted for the targeted receptors. Disruption of *Adora2A*

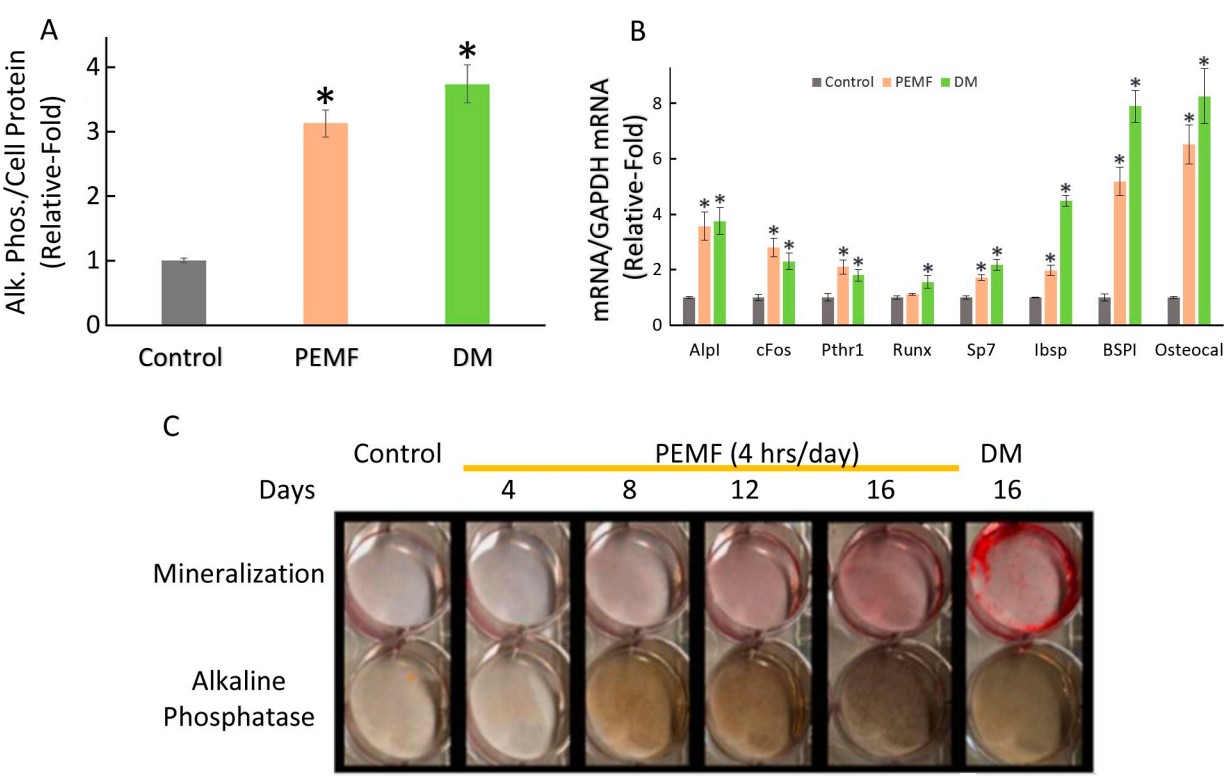

**Fig 1. PEMF induces osteoblast differentiation.** A. Relative alkaline phosphate enzymatic activity normalized to cell protein after 4 hours daily treatment with PEMF for 12 days or differentiation medium (DM) for 12 days. * indicates p<0.05, Student's t-test compared to untreated control. B. Relative expression of osteoblast differentiation genes as indicated, normalized to *GAPDH* after 4 hours/day treatment with PEMF for 12 days or DM for 12 days. * indicates p<0.05, Student's t-test compared to untreated control of same gene. C. Time series of Alizarin red staining for mineralization and alkaline phosphate activity staining after PEMF or DM treatment.

resulted in almost no significant difference in its enzymatic activity and only a delayed induction of alkaline phosphatase overexpression upon PEMF stimulation compared to the WT cells up to 16 days as measured by enzymatic assay, qRT-PCR and cell staining (**Fig 3A–3C**). On the other hand, disruption of *Adora3* resulted in a reduced response compared to the WT cells as measured by both *Alp* in protein and mRNA levels throughout the stimulation periods. Incidentally, treatment with DM for 16 days showed a similar pattern of more prominent loss of alkaline phosphatase activation in *Adora3* or doubly disrupted ($A_{2A}/A_3$) cells, but not in *Adora2A* disrupted in MC3T3-E1 (**Fig 3B**). For PEMF-mediated activation of *Alp* expression, a similar dependence was observed for *Adora 3*, but not *Adora 2A*, as measured by alkaline phosphatase staining (**Fig 3C**), and by Alizarine Red staining (**Fig 3D**).

## PEMF-induced *cFos* and *Pthr1* expression is blunted in *Adora2A* disruption

Next we went on to investigate the roles of the $A_{2A}$ and $A_3$ receptors in PEMF-mediated activation of other osteoblast differentiation genes, viz. *cFos*, *Ibsp*, *Pthr1*, *Runx2* and *Sp7*. In the wild-type MC3T3-E1 cells, all these genes showed increased expression up to 12–16 days of PEMF stimulation (**Fig 4A–4E**). DM-mediated increases in *Ibsp*, *Runx2* and *Sp7* were also observed, however for *cFos* and *Pthr1* the activation was very modest or non-existent. After *Adora2A* disruption, a general tendency of dramatically blunted response to PEMF stimulation was observed for both *cFos* and *Pthr1* gene expression, whereas in the case of *Adora3* disruption,

**1. Donor Plasmids with homology arms**

**2. gRNAs Cloned Cas-9 expressing Vector**

Cotransfection

**Fig 2. *Adora 2A* and *3* disruption by CRISPR-Cas9.** MC3T3-E1 cells were co-transfected with a pUC donor plasmid with RFP-PGK-BSD or GFP-PGK-Puro flanked by homolog arms (1) and pCas-Guide plasmid containing Cas9 and gRNA corresponding to cleavage site in the first exon in each gene (2). Expected insertions of the cassette after homology-dependent recombination (HDR) are shown at the bottom.

the effect on expression of those genes was less remarkable. On the other hand, PEMF stimulation was more blunted in *Ibsp* and *Sp7* due to *Adora3* disruption, but not *Adora2* disruption; while *Runx2* activation seemed to be independent of either.

## PEMF-induced expression of differentiation genes and the effect of *Adora* disruption on their message levels are also seen at the protein level

To correlate the molecular data at the messenger RNA level with protein expression of osteoblast markers, effects of PEMF on the mature osteoblast markers BSP, IBSP, and osteocalcin in wild-type and *Adora*-disrupted MC3T3-E1 cells was evaluated by Western Blot (Fig 5). Normalized to GAPDH expression serving as a loading control, both PEMF and DM exposure induced increased expression of all the above proteins compared to control cells in the wild-type and *Adora2* disrupted cells, but not in *Adora3* or doubly disrupted cells.

## *Adora3* reconstitution in *Adora3* disrupted cells restores alkaline phosphatase activation by PEMF

To address the specificity of gene disruption, and to eliminate the possibility of off-target effects masquerading as an $A_3$-specific result, we took the *Adora3* disrupted MC3T3-E1 cells and "knocked-in" the *Adora3* ORF into the ROSA26 locus under the control of a cumate-inducible promoter. When Adora3 function was thus reconstituted in *Adora3* disrupted cells, PEMF response as determined through *Alp* gene activation was found to be restored close to

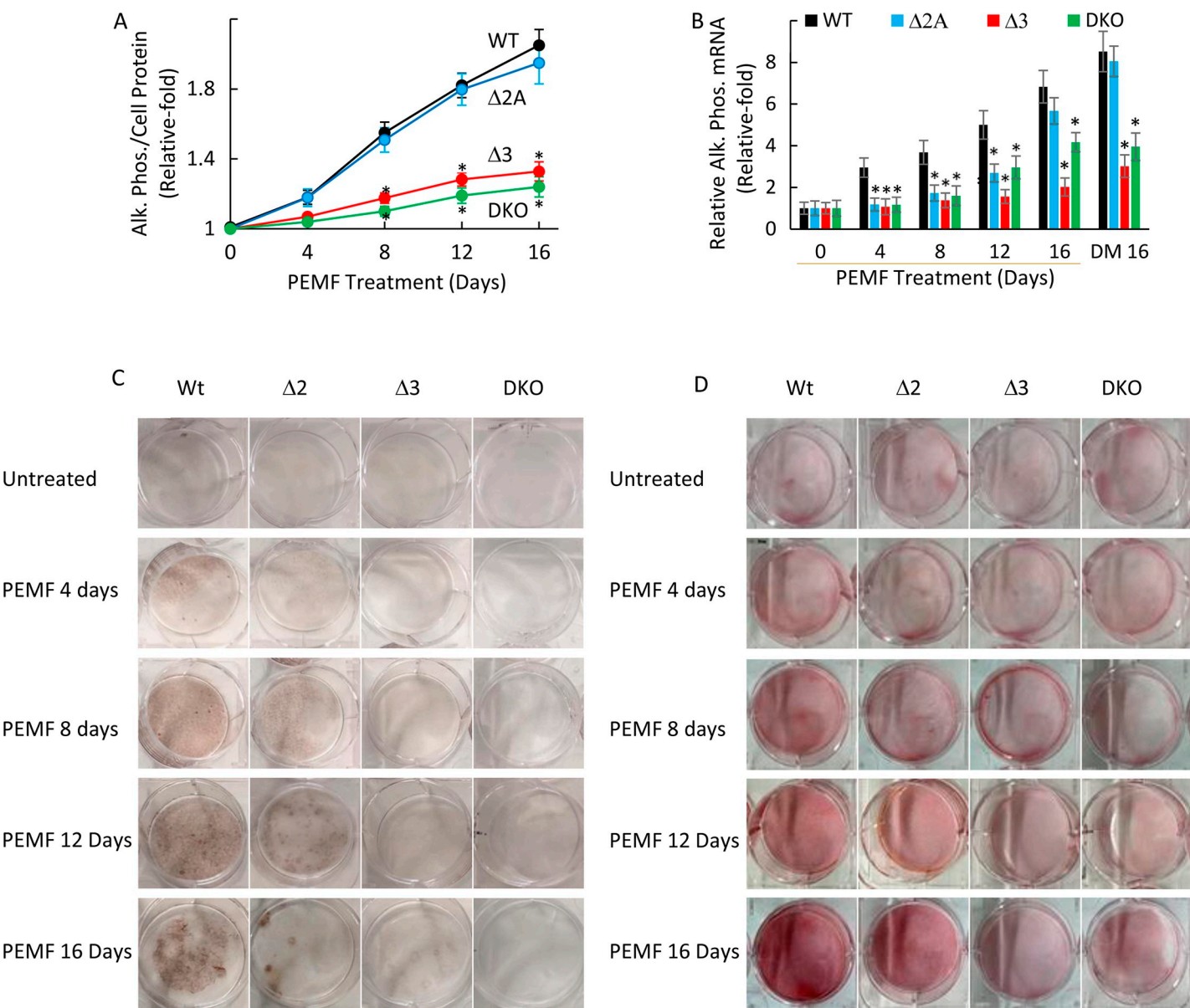

**Fig 3. *Adora3* disruption affects PEMF activation of alkaline phosphatase.** A. Relative alkaline phosphate enzymatic activity normalized to cell protein after 4 hours daily treatment with PEMF for 4, 8, 12 and 16 days. * indicates $p < 0.05$, Student's t-test compared to WT control for each time-point. B. Relative expression of alkaline phosphatase gene normalized to GAPDH after 4 hours daily treatment with PEMF up to 16 days or DM for 16 days. * indicates $p < 0.05$, Student's t-test compared to WT for each time-point. C. Alkaline phosphatase staining of the MC3T3-E1 cells without and with PEMF treatment. D. Alizarin Red staining of the MC3T3-E1 cells without and with PEMF treatment.

the typical response level observed in the wild-type cells (**Fig 6**). Expression of *cFos* and *Pthr1* mRNA remained unaltered either as a result of *Adora3* disruption or *Adora3* reconstitution.

## Discussion

Though PEMF has been known to stimulate non-union fracture healing for more than four decades [3,4,7], its effect on the initial step of wound healing, inflammation, and oxidative stress has been investigated only recently in terms modulation of plasma membrane $Ca^{2+}$, ATP/ATPase or heat shock responses [18–21]. Multiple studies examining the pre-osteoblast

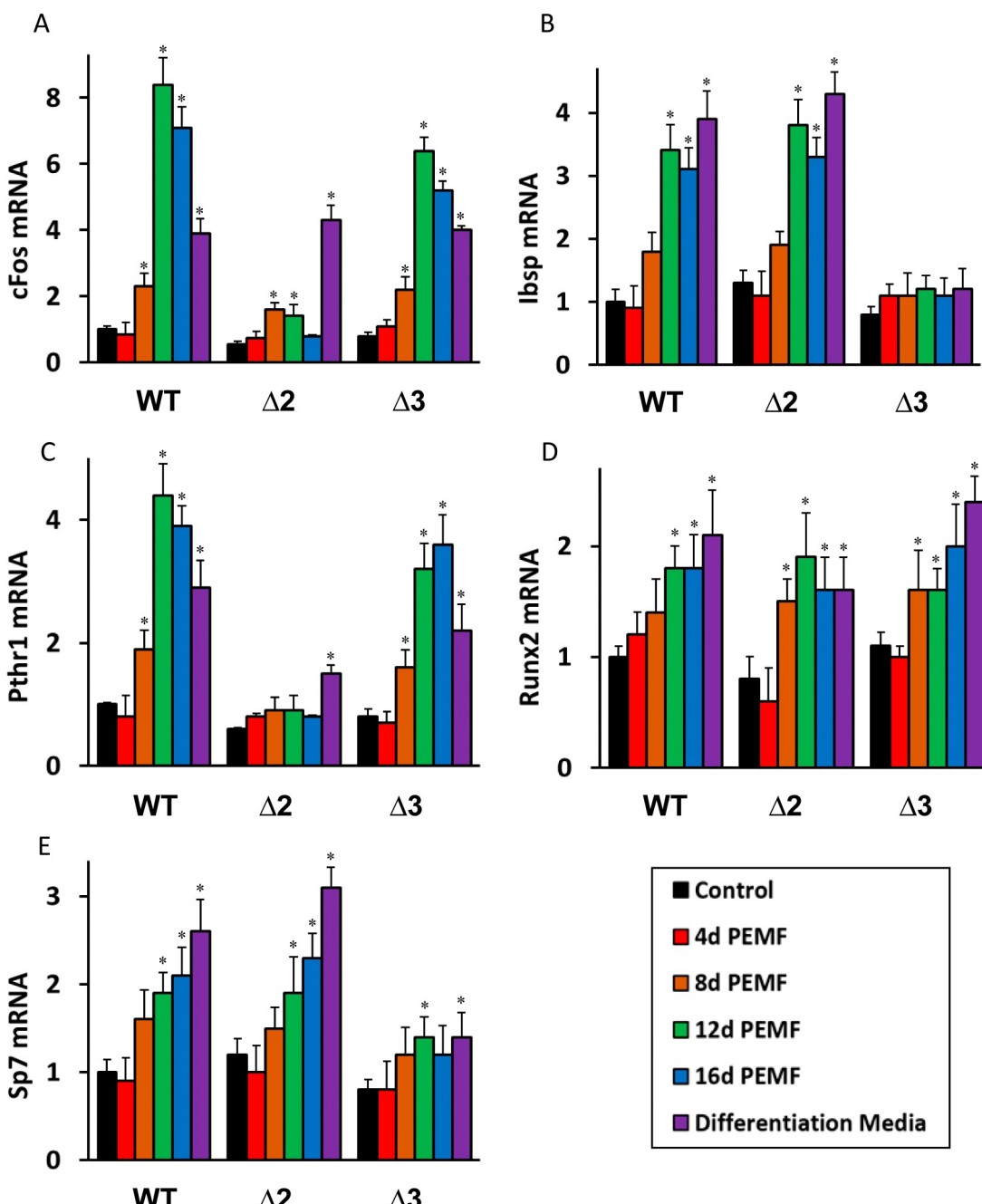

**Fig 4.** *Adora2* and *Adora3* disruption affects PEMF activation of *cFos*, *Ibsp*, *Pthr1*, *Runx2* and *Sp7* (A-E). Relative fold-induction of each gene expression as measured by qRT-PCR with *Gapdh* as standard, with PEMF treatment up to 16 days (4 hours daily) or DM for 16 days. * indicates $p < 0.05$, Student's t-test compared to untreated control at day 8 of same gene of same cell-type.

to osteoblast transition and differentiation have focused on a core set of genes that include alkaline phosphatase (*Alp*), cellular proto-oncogene derived from the Finkel–Biskis–Jinkins murine osteogenic sarcoma virus (*cFos*), integrin-binding sialoprotein (*Ibsp*), parathyroid hormone 1 receptor (*Pthr1*), Runt-related transcription factor 2 (*Runx2*) and transcription factor *Sp7*, also called Osterix (*Sp7/Osterix*) [33–36]. PEMF has been shown to activate ALP, insulin-like growth factor 1, collagen type 1, osteocalcin and Runx2 expression in a $Ca^{2+}$-dependent

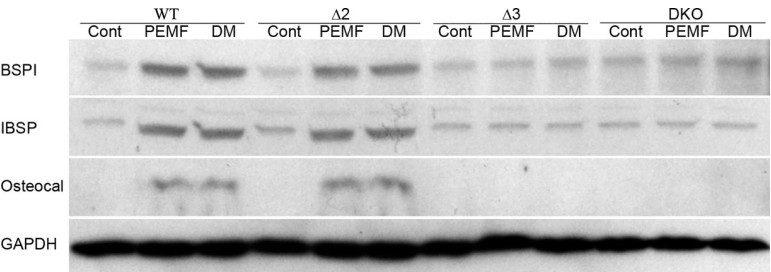

**Fig 5.** *Adora3* **disruption affects PEMF mediated overexpression of BSP, IBSP and Osteocalcin.** Wild-type (WT), *Adora2* (Δ2), *Adora3* (Δ3) and Double Knock-Out (DKO) MC3T3-E1 cells were treated with PEMF (4 hours daily) or differentiation medium (DM) for 12 days and Western Blots for fractionated whole-cell proteins were probed with each of the indicated antibodies.

fashion [15,37]; however, the upstream signaling pathways leading to these activations are poorly understood. Prior studies mentioned above and others have shown that the only adenosine receptors that are clearly modulated by PEMF were the $A_{2A}$ and $A_3$ ADRs [22,29,38–42]. The present study explores a more detailed profiling of PEMF-responsive genes, and focuses on the role of a subset of adenosine receptors in PEMF-mediated osteoblast differentiation.

Previous studies showed a connection between PEMF-mediated effects on proliferation and inflammation in multiple cell types through adenosine receptors, where specifically $A_{2A}$ and $A_3$ expression were amplified and proteins migrated to the plasma membrane [22,29,42]. Furthermore, in separate contexts, these receptors, $A_{2A}$ and $A_3$ were shown to be involved in osteoblast differentiation under non-PEMF physiological conditions [24,43]. These two separate but complementary observations in turn led us to test the hypothesis whether these

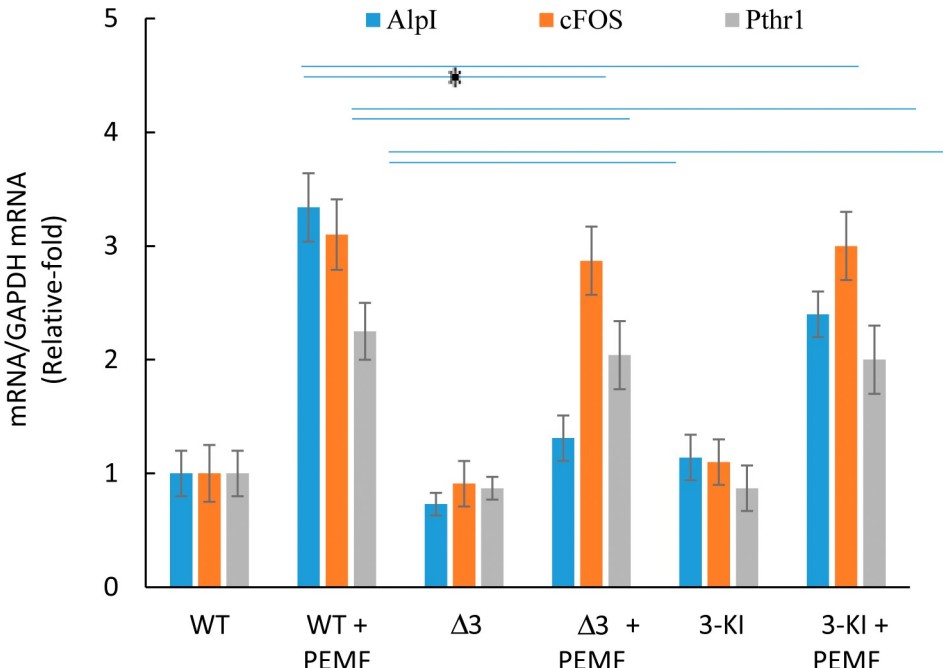

**Fig 6.** *Adora3* **knock-in in** *Adora3* **disrupted cells restores PEMF-mediated induction of alkaline phosphatase in PEMF-treated cells.** Expressions of alkaline phosphate, *cFos* and *Pthr1* genes were measured by qRT-PCR with *Gapdh* as standard with PEMF treatment for 8 days. * indicates $p < 0.05$, Student's t-test compared to PEMF treatment in WT cells.

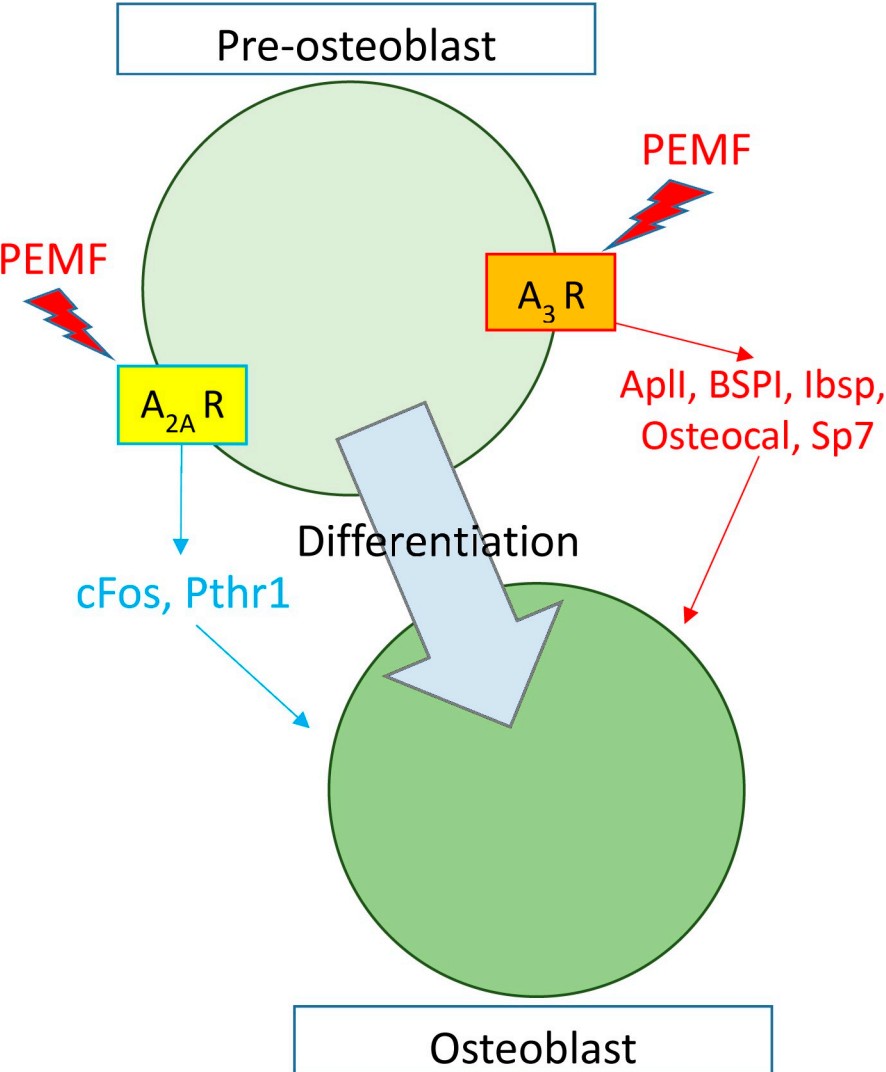

**Fig 7. Schematic diagram role of Adenosine Receptors in PEMF-mediated MC3T3-E1 differentiation.** PEMF stimulated osteoblastic differentiation depends on Adenosine Receptor $A_{2A}$ for cFos and Pthr1 overexpression, and $A_3$ for AlpI activation.

receptors are modulators of osteoblast differentiation under PEMF stimulation conditions as well. Using CRISPR-Cas9-based gene editing techniques to disrupt these two genes, individually or in combination, we were able to show functional implications of *Adora2A* and *Adora3* in PEMF-mediated osteoblast differentiation. We kept our study focused mainly on mRNA expression of differentiation genes because it has been reported in several previous studies that their protein expressions correlated mRNA expression [44–46], which also matched our observation in the case of AlpI enzymatic assay (**Fig 3**). Our results indicate that the $A_{2A}$ and $A_3$ receptors are required for at least two parallel and complementary signaling pathways during the differentiation process: $A_{2A}$ seems to be important in PEMF-mediated stimulation of *cFos* and *Pthr1* pathways while $A_3$ appears to modulate *Alp*, *BSPI*, *Ibsp*, *Osteocalcin* and *Sp7* pathways (summarized in **Fig 7**). The messenger RNA data determined by qRT-PCR was confirmed at the protein level for the key osteoblast differentiation protein markers. Curiously, a similar pattern of gene modulation was observed under differentiation using a chemically

defined medium (DM), which indicates a commonality of molecular pathways being activated by these two disparate stimuli. One plausible scenario is that PEMF, like DM, activates release of certain ligands that modulate the activity of these receptors. This possibility is currently under investigation.

To exclude the possibility that the loss of function observed in *Adora3* gene disrupted cells was due to off-target effects, "knock-in" experiments were performed to reintroduce the *Adora3* ORF into the open chromatin *ROSA26* locus under control of the inducible cumate promoter in *Adora3*-disrupted MC3T3-E1 cells. These reconstituted $A_3$ cells showed a PEMF response pattern similar to that observed for the wild-type pre-osteoblast MC3T3-E1 cells, which confirmed the specificity of the $A_3$ receptor's role in Alp pathway activation by both PEMF and DM.

This study utilized a targeted approach focusing on a subset of adenosine receptors that were shown to be modulated by PEMF during osteoblast differentiation, and to behave as facilitators of PEMF action in the initial phases of osteoblast differentiation. This modulation was observed to operate through at least two parallel paths. We do not envision these receptors as direct PEMF-sensing receptors; however they do appear to respond to upstream PEMF-initiated signals. In order to identify PEMF receptors, key PEMF-signaling node genes and downstream PEMF-responsive genes, an untargeted approach can be envisioned where genome-wide gene silencing and screening can be carried out using a knock-out library and reporter system followed by informatics analysis to identify the most upstream PEMF sensing genes. This strategy is currently being explored.

## Supporting information

**S1 Raw images. Complete unedited immunoblots of transferred proteins from gel fractionation.**
(PDF)

**S1 File.**
(PDF)

## Author Contributions

**Conceptualization:** Joseph A. DiDonato.

**Data curation:** Niladri S. Kar, Joseph A. DiDonato.

**Formal analysis:** Niladri S. Kar, Daniel Ferguson, Joseph A. DiDonato.

**Funding acquisition:** Joseph A. DiDonato.

**Investigation:** Niladri S. Kar, Daniel Ferguson, Joseph A. DiDonato.

**Methodology:** Niladri S. Kar, Joseph A. DiDonato.

**Project administration:** Niladri S. Kar, Nianli Zhang, Erik I. Waldorff, James T. Ryaby, Joseph A. DiDonato.

**Resources:** Niladri S. Kar.

**Supervision:** Niladri S. Kar, Nianli Zhang, James T. Ryaby, Joseph A. DiDonato.

**Validation:** Niladri S. Kar, Joseph A. DiDonato.

**Visualization:** James T. Ryaby, Joseph A. DiDonato.

**Writing – original draft:** Niladri S. Kar, Joseph A. DiDonato.

**Writing – review & editing:** Niladri S. Kar, Daniel Ferguson, Nianli Zhang, Erik I. Waldorff, James T. Ryaby, Joseph A. DiDonato.

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
