## [Decision Letter · Decision Letter 0]

24 Jul 2020

PONE-D-20-20625

Pulsed-electromagnetic-field induced osteoblast differentiation requires activation of genes downstream of adenosine receptors A2A and A3

PLOS ONE

Dear Dr. DiDonato,

Thank you for submitting your manuscript to PLOS ONE. After careful consideration, we feel that it has merit but does not fully meet PLOS ONE’s publication criteria as it currently stands. Therefore, we invite you to submit a revised version of the manuscript that addresses the points raised during the review process.

This study has been performed without taking into consideration of a fundamental, basic scientific point: in order to put in comparison data they must belong to experiments performed at the same time point of the same experiments. To put in comparison data belonging to different time points is a great mistake. Therefore other experiments must be performed and all must be done from the beginning in order to give obtain comparable data.

This Editor gives to the Authors this unique opportunity and asks to them to answer to the other criticisms raised by both the referees.

We look forward to receiving your revised manuscript.

Kind regards,

Gianpaolo Papaccio, M.D., Ph.D.

Academic Editor

PLOS ONE

Journal Requirements:

2. Thank you for including your competing interests statement; "

NSK, DF, and JAD have no conflicts to declare. NZ, EIW, and JTR are employed by and own stock in Orthofix Medical Inc."

We note that one or more of the authors have an affiliation to the commercial funders of this research study : Orthofix Medical Inc

Reviewers' comments:

Reviewer's Responses to Questions

**Comments to the Author**

1. Is the manuscript technically sound, and do the data support the conclusions?

Reviewer #1: Partly

Reviewer #2: Yes

2. Has the statistical analysis been performed appropriately and rigorously? 

Reviewer #1: Yes

Reviewer #2: Yes

3. Have the authors made all data underlying the findings in their manuscript fully available?

Reviewer #1: Yes

Reviewer #2: Yes

4. Is the manuscript presented in an intelligible fashion and written in standard English?

Reviewer #1: Yes

Reviewer #2: No

5. Review Comments to the Author

Reviewer #1: In this paper Authors evaluated the effect of Pulsed-electromagnetic-field (PEMF) treatment on the mouse preosteoblasts differentiation and identified adenosine receptor A3 as the main responsible.

The paper is interesting but there are some points that should be addressed.

Authors should describe results in a detached way and move the comments into the discussion.

Authors should add in M&M section the indication of time points at which experiments were performed.

Why did Authors compare mineralization and Alkaline phosphatase of the two groups (PEMF and DM) at different time points (4, 8, 12 and 16 days for PEMF and only 16 for DM)? Moreover, they talk about experiments carried out for up to 16 days but in the figure 1C they also showed 20 days for PEMF treatment. Please, correct it.

Authors should present histograms with the same layout.

Authors should confirm gene expression also considering the relative protein production.

Reviewer #2: The manuscript is interesting. The authors aimed to investigate the effect of PEMF on osteoblast differentiation of mouse preosteoblasts and demonstrated that the Adora3 open reading frame was inserted into the ROSA26 locus in Adora3 disrupted cells culminating in rescued PEMF responsiveness. Although this, there some concerns that need to be addressed.

The authors must reorganize the results especially in terms of times. Each experiment is performed at different times. Times must be the same in order to compare the results. The mature osteoblast expresses osteopontin, Bone sialo-protein, and osteocalcin. Therefore, the authors must evaluate the expression of these markers. Moreover, molecular data must be confirmed by corresponding protein expression.

6. PLOS authors have the option to publish the peer review history of their article (what does this mean?). If published, this will include your full peer review and any attached files.

Reviewer #1: No

Reviewer #2: No

---

## [Author Response · Author response to Decision Letter 0]

27 Jan 2021

Reviewer #1: In this paper Authors evaluated the effect of Pulsed-electromagnetic-field (PEMF) treatment on the mouse preosteoblasts differentiation and identified adenosine receptor A3 as the main responsible.

The paper is interesting but there are some points that should be addressed.

Authors should describe results in a detached way and move the comments into the discussion.

Authors should add in M&M section the indication of time points at which experiments were performed.

Why did Authors compare mineralization and Alkaline phosphatase of the two groups (PEMF and DM) at different time points (4, 8, 12 and 16 days for PEMF and only 16 for DM)? Moreover, they talk about experiments carried out for up to 16 days but in the figure 1C they also showed 20 days for PEMF treatment. Please, correct it.

Authors should present histograms with the same layout.

Authors should confirm gene expression also considering the relative protein production.

Reviewer #2: The manuscript is interesting. The authors aimed to investigate the effect of PEMF on osteoblast differentiation of mouse preosteoblasts and demonstrated that the Adora3 open reading frame was inserted into the ROSA26 locus in Adora3 disrupted cells culminating in rescued PEMF responsiveness. Although this, there some concerns that need to be addressed.

The authors must reorganize the results especially in terms of times. Each experiment is performed at different times. Times must be the same in order to compare the results. The mature osteoblast expresses osteopontin, Bone sialo-protein, and osteocalcin. Therefore, the authors must evaluate the expression of these markers. Moreover, molecular data must be confirmed by corresponding protein expression.

Response to Reviewers:

Reviewer #1: In this paper Authors evaluated the effect of Pulsed-electromagnetic-field (PEMF) treatment on the mouse preosteoblasts differentiation and identified adenosine receptor A3 as the main responsible.

The paper is interesting but there are some points that should be addressed.

Authors should describe results in a detached way and move the comments into the discussion.

• As suggested, the Results section has been rewritten in a more detached way, and the irrelevant materials have been moved to the Discussion section as requested. 

Authors should add in M&M section the indication of time points at which experiments were performed.

• Time-frames for each experiment are now explicitly mentioned in the Methods sections, and, together with graph colors scheme, made more consistent amongst the different panels/figures.

Why did Authors compare mineralization and Alkaline phosphatase of the two groups (PEMF and DM) at different time points (4, 8, 12 and 16 days for PEMF and only 16 for DM)? Moreover, they talk about experiments carried out for up to 16 days but in the figure 1C they also showed 20 days for PEMF treatment. Please, correct it.

Authors should present histograms with the same layout.

• In the revised manuscript we also now have examined both PEMF and Differentiation media (DM) at the same timepoints, and, together with graph colors scheme, we made made more consistent amongst the different panels/figures.

Authors should confirm gene expression also considering the relative protein production.

• Protein expression as depicted by Western Blot, as a confirmatory data for RT-PCR data, have now been included in the new Figure 5 as requested by the reviewer.

Reviewer #2: The manuscript is interesting. The authors aimed to investigate the effect of PEMF on osteoblast differentiation of mouse preosteoblasts and demonstrated that the Adora3 open reading frame was inserted into the ROSA26 locus in Adora3 disrupted cells culminating in rescued PEMF responsiveness. Although this, there some concerns that need to be addressed.

The authors must reorganize the results especially in terms of times. Each experiment is performed at different times. Times must be the same in order to compare the results. 

• In the original submission, single time-frames in the initial submission’s experiments were exploratory in nature and later were expanded as a time-series (dose response) over a wider temporal range. In the revised manuscript, as mentioned above for Reviewer 1’s request, we have changed each of the experimental time-points to make them consistent among the different panels/figures.

The mature osteoblast expresses osteopontin, Bone sialo-protein, and osteocalcin. Therefore, the authors must evaluate the expression of these markers. Moreover, molecular data must be confirmed by corresponding protein expression.

• Both messenger RNA (Figure 1B) and protein expression (Figure 5) are now included for the mature osteoblast markers, Osteocalcin and Osteopontin, as requested. As the original manuscript included qRT-PCR data on Bone sialo-protein II (Ibsp), the corresponding protein expression data has also been added (Figure 5) as well.

---

## [Decision Letter · Decision Letter 1]

11 Feb 2021

Pulsed-electromagnetic-field induced osteoblast differentiation requires activation of genes downstream of adenosine receptors A2A and A3

PONE-D-20-20625R1

Dear Dr. DiDonato,

We’re pleased to inform you that your manuscript has been judged scientifically suitable for publication and will be formally accepted for publication once it meets all outstanding technical requirements.

Kind regards,

Gianpaolo Papaccio, M.D., Ph.D.

Academic Editor

PLOS ONE

Additional Editor Comments (optional):

Reviewers' comments:

Reviewer's Responses to Questions

**Comments to the Author**

1. If the authors have adequately addressed your comments raised in a previous round of review and you feel that this manuscript is now acceptable for publication, you may indicate that here to bypass the “Comments to the Author” section, enter your conflict of interest statement in the “Confidential to Editor” section, and submit your "Accept" recommendation.

Reviewer #1: All comments have been addressed

Reviewer #2: All comments have been addressed

2. Is the manuscript technically sound, and do the data support the conclusions?

Reviewer #1: (No Response)

Reviewer #2: Yes

3. Has the statistical analysis been performed appropriately and rigorously? 

Reviewer #1: (No Response)

Reviewer #2: Yes

4. Have the authors made all data underlying the findings in their manuscript fully available?

Reviewer #1: (No Response)

Reviewer #2: Yes

5. Is the manuscript presented in an intelligible fashion and written in standard English?

Reviewer #1: (No Response)

Reviewer #2: Yes

6. Review Comments to the Author

Reviewer #1: (No Response)

Reviewer #2: (No Response)

7. PLOS authors have the option to publish the peer review history of their article (what does this mean?). If published, this will include your full peer review and any attached files.

Reviewer #1: No

Reviewer #2: No

---

## [Editor Report · Acceptance letter]

17 Feb 2021

PONE-D-20-20625R1 

Pulsed-electromagnetic-field induced osteoblast differentiation requires activation of genes downstream of adenosine receptors A2A and A3 

Dear Dr. DiDonato:

I'm pleased to inform you that your manuscript has been deemed suitable for publication in PLOS ONE. Congratulations! Your manuscript is now with our production department. 

Kind regards, 

on behalf of

Prof. Gianpaolo Papaccio 

Academic Editor

PLOS ONE